# Interpreting roles of mutations associated with the emergence of *S. aureus* USA300 strains using transcriptional regulatory network reconstruction

**Saugat Poudel[1], Jason Hyun[1], Ying Hefner[1], Jon Monk[2], Victor Nizet[3,4], Bernhard O Palsson[1,3,4]***

[1]University of California, San Diego, La Jolla, United States; [2]Palmona Pathogenomics, Menlo Park, United States; [3]Collaborative to Halt Antibiotic-Resistant Microbes (CHARM), Department of Pediatrics, University of California San Diego, La Jolla, United States; [4]Department of Pediatrics, University of California San Diego, La Jolla, United States

*For correspondence:
palsson@ucsd.edu

**Competing interest:** The authors declare that no competing interests exist.

## eLife assessment

This study presents **valuable** findings on core genome mutations that might have driven the emergence of the *Staphylococcus aureus* lineage USA300, a frequent cause of community-acquired infections. The authors present a **solid** novel approach that combines genome-wide association studies and RNA-expression analyses, both applied to extensive publicly available datasets. This approach generated an intriguing hypothesis that should be validated experimentally. The work will interest microbiologists working in genomic epidemiology and phenotype-genotype association studies.

**Abstract** The *Staphylococcus aureus* clonal complex 8 (CC8) is made up of several subtypes with varying levels of clinical burden; from community-associated methicillin-resistant *S. aureus* USA300 strains to hospital-associated (HA-MRSA) USA500 strains and ancestral methicillin-susceptible (MSSA) strains. This phenotypic distribution within a single clonal complex makes CC8 an ideal clade to study the emergence of mutations important for antibiotic resistance and community spread. Gene-level analysis comparing USA300 against MSSA and HA-MRSA strains have revealed key horizontally acquired genes important for its rapid spread in the community. However, efforts to define the contributions of point mutations and indels have been confounded by strong linkage disequilibrium resulting from clonal propagation. To break down this confounding effect, we combined genetic association testing with a model of the transcriptional regulatory network (TRN) to find candidate mutations that may have led to changes in gene regulation. First, we used a De Bruijn graph genome-wide association study to enrich mutations unique to the USA300 lineages within CC8. Next, we reconstructed the TRN by using independent component analysis on 670 RNA-sequencing samples from USA300 and non-USA300 CC8 strains which predicted several genes with strain-specific altered expression patterns. Examination of the regulatory region of one of the genes enriched by both approaches, *isdH*, revealed a 38-bp deletion containing a Fur-binding site and a conserved single-nucleotide polymorphism which likely led to the altered expression levels in USA300 strains. Taken together, our results demonstrate the utility of reconstructed TRNs to address the limits of genetic approaches when studying emerging pathogenic strains.

## Introduction

Comparative genomic methods represent an important approach to understand the emergence and evolution of new strains of pathogens. In *Staphylococcus aureus* alone, whole-genome comparisons have enabled rapid characterization of genetic basis for antibiotic resistance, increased virulence, host specificity, and altered metabolic capabilities (*Young et al., 2019*; *Bosi et al., 2016*; *Choudhary et al., 2018*; *Correction for Copin, 2019*; *Krishna et al., 2018*). However, genome-wide linkage disequilibrium and strong population structure currently limit the differentiation of causative alleles from genetically linked ones. By calculating lineage level associations, methods like bugwas address these issues for single, recurring phenotypes like antibiotic resistance (*Earle et al., 2016*).

Emerging clonal complexes, on the other hand, exhibit multiple complex phenotypes that may contribute to their emergence and proliferation. For example, USA300 strains carry antibiotic resistance cassettes, Panton–Valentine Leukocidin (PVL) associated with pyogenic skin infections, increased ability to colonize locations outside of the nasopharynx, etc. (*Young et al., 2019*; *Diep et al., 2008*; *Faden et al., 2010*). As these strains often emerge clonally from closely related 'ancestral strains', efforts to discern causal mutations that lead to their increased clinical burden is hampered by strong population-stratification and genome-wide linkage disequilibrium (*Steinig et al., 2019*; *Challagundla et al., 2018b*; *Bal et al., 2016*). Though recombination at species level is common in *S. aureus*, within clade recombination rates tend to be lower, thus preserving the linkage between mutations (*Challagundla et al., 2018b*; *Uhlemann et al., 2014*; *Challagundla et al., 2018a*; *Everitt et al., 2014*). Due to this limitation, studies of emerging strains often focus on gene-level analysis such as acquisition of mobile genetic elements or loss of gene function as their effect on phenotype is easier to determine than that of point mutations (*Thurlow et al., 2012*; *Boyle-Vavra et al., 2015*). Computational modeling methods can help tackle these challenges by predicting phenotype differences between strains, thus acting as a sieve to filter enriched mutations with potential phenotypic effects and therefore find candidate causal mutations (*Nishizaki et al., 2020*; *Choi et al., 2012*; *Kavvas et al., 2020*). Even if experimentally intractable, the large possible phenotypic space of an organism can be explored quickly with computational models. Taking advantage of these modeling techniques, we use a reconstruction of the transcriptional regulatory network (TRN) of clonal complex 8 (CC8) strains to find USA300-specific mutations that are associated with changes in gene regulation.

First, we used De Bruijn graph genome-wide association study (DBGWAS) to discover enriched mutations associated with the USA300 strain within CC8 (*Jaillard et al., 2018*). Due to clonal expansion of USA300 strains from their progenitors within CC8, the enriched USA300-specific mutations were in high linkage disequilibrium. Further complicating the matter, we found that almost all mutations enriched within open reading frames (ORFs) were unique to USA300 lineage and not found in any other clonal complexes, precluding identification of potential causative mutations by homoplasy. Instead, we turned to reconstruction of a TRN to identify genes that were both associated with an enriched mutation and had altered regulation in USA300 strains. We built an independent component analysis (ICA)-based reconstruction of the TRN using 670 publicly available RNA-sequencing samples from both USA300 and non-USA300 CC8 strains. By factoring the RNA-sequencing data into a series of signals and their activities, the ICA-based reconstruction of the TRN shows both the static gene–regulator interaction and the dynamic activity of these interactions in a sample-specific manner (*Sastry et al., 2019*). However, ICA is a generalized signal extraction algorithm and therefore does not distinguish between biological sources of signals like regulatory elements and 'artificial' sources that can be created by sourcing data from multiple strains. Therefore, in addition to signals associated with gene regulators, ICA also outputs signals associated with strain-specific changes in gene regulation. Furthermore, by utilizing RNA-sequencing data from hundreds of samples to identify genes with strain-specific expression patterns, this approach is more likely to find strain-specific differences than previous approaches that focus on specific conditions (*Jones et al., 2014*; *Iqbal et al., 2016*).

This analysis revealed several genes with distinct expression patterns in USA300 strains that were also associated with DBGWAS-enriched mutations. One of these genes, *isdH*, which encodes a haptoglobin-binding protein, showed several enriched mutations in the regulatory region of USA300 strains including the deletion of the Fur repressor-binding site. Additionally, the isdH gene generally had higher expression levels in samples from USA300 strains, connecting the mutation enriched by DBGWAS with differences in TRN enriched by ICA. Overall, our analysis shows how the reconstruction

of TRN can be used to extend the limits of current GWAS approaches when studying emerging populations of bacterial pathogens.

## Results

### Classifying USA300 and non-USA300 genomes based on genetic markers

We sought to compare the genetic differences between USA300 community-associated methicillin-resistant *S. aureus* (CA-MRSA) strains and other subtypes within CC8 that have lower clinical and community burden. Given that both subtypes exist within the same clonal complex, this comparison allowed us to probe the genetic basis for the success of USA300 strains with limited confounding effects of different genetic backgrounds. We downloaded 2033 *S. aureus* genomes for analysis and excluded six of them with genome length of less than 2.5 million base pairs. The CC8 pangenome consisted of 19176 gene clusters with 2291 core genes that were present in at least ~95% of the genomes analyzed. Among the remainder of the genes, 931 were categorized as accessory genes and 15,954 were found in less than 5% of the genomes (*Figure 1—figure supplement 1A*). The collection formed a closed pangenome, as adding new genomes did not introduce many new genes (*Figure 1A*), suggesting that our collection had a good gene-level coverage of the CC8 pangenome. We confirmed the pangenome coverage with Roary (*Figure 1—figure supplement 1B*). To get a higher resolution view of these genomes, we surveyed unique alleles within the ORFs and in the 300-bp 5′ upstream and 3′ downstream sequences. We found a larger number of mutations within the ORFs, indicating the presence of greater genetic variation in the ORFs than in the neighboring regulatory regions. This is reflective of the fact that most of *S. aureus* genome sequence comprises of ORFs for example ~84% of TCH1516 genome is part of an ORF.

Next, we classified the CC8 genomes into USA300 and non-USA300 strains using Genetic Marker Inference (GMI). GMI was previously developed to rapidly and systematically identify different subclades within inner-CC8 strictly based on genetic markers (*Bowers et al., 2018*). In this scheme, USA300 genomes can be differentiated from non-USA300 CC8 genomes by the presence of either SCC*mec* IVa or the presence of PVL in case of methicillin-susceptible *S. aureus* (MSSA).

To identify the root of the USA300 clades, we first traversed up nodes of the phylogenetic tree starting from known USA300 strain TCH1516 and determined the number of strains, fraction PVL positive and fraction SCC*mec* IVa positive for each node during traversal. The root was placed at the last node where >90% of the strains within the subclade represented by the daughter nodes were SCC*mec* IVa and PVL positive (*Figure 1B*). As phylogenetic trees are nested, root finding with this procedure is not dependent on the starting USA300 strain. Same root was identified when the procedure was initialized with another well-known USA300 reference strain FPR3757 (*Figure 1—figure supplement 1C*). Combining the genetic markers with phylogenetic grouping led to the classification of 1449 genomes as USA300 and 589 genomes as non-USA300 (*Figure 1C*, *Supplementary file 1*). Strains previously identified as 'early USA300' were not part of our USA300 classification (*Bowers et al., 2018*). While many of these strains are PVL positive, they have variable SCC*mec* types and therefore are likely to be clinically distinct from the USA300 strains.

### Enriching USA300-specific genes and mutations using DBGWAS

After classifying the genomes into USA300 and non-USA300 strains, we identified genes and mutations associated with each subtype by using the DBGWAS (*Jaillard et al., 2018*). We used 2030 genomes for this analysis; the 2027 genomes in pangenomics analysis above were 'spiked' with three well-known CC8 genomes – TCH1516, COL, and Newman – to help annotate the DBGWAS unitigs. DBGWAS provides a reference genome-free method for conducting GWAS analysis in prokaryotes by building a compacted De Brujin graph to represent the pan genome of input sequences. The nodes of the graph represent unique compacted k-mers that are joined by edges to other nodes with k-mers that appear adjacent to it in genomes. The procedure enriches unique k-mers that appear with different frequencies in each classification and outputs the enriched k-mer as well as its genetic neighborhood (called 'components') from the De Bruijn graph. Visualizing the components associated with the enriched k-mers makes it easier to interpret the k-mers and makes it easy to identify large structural variations (e.g. cassette acquisition) which are often represented by multiple enriched

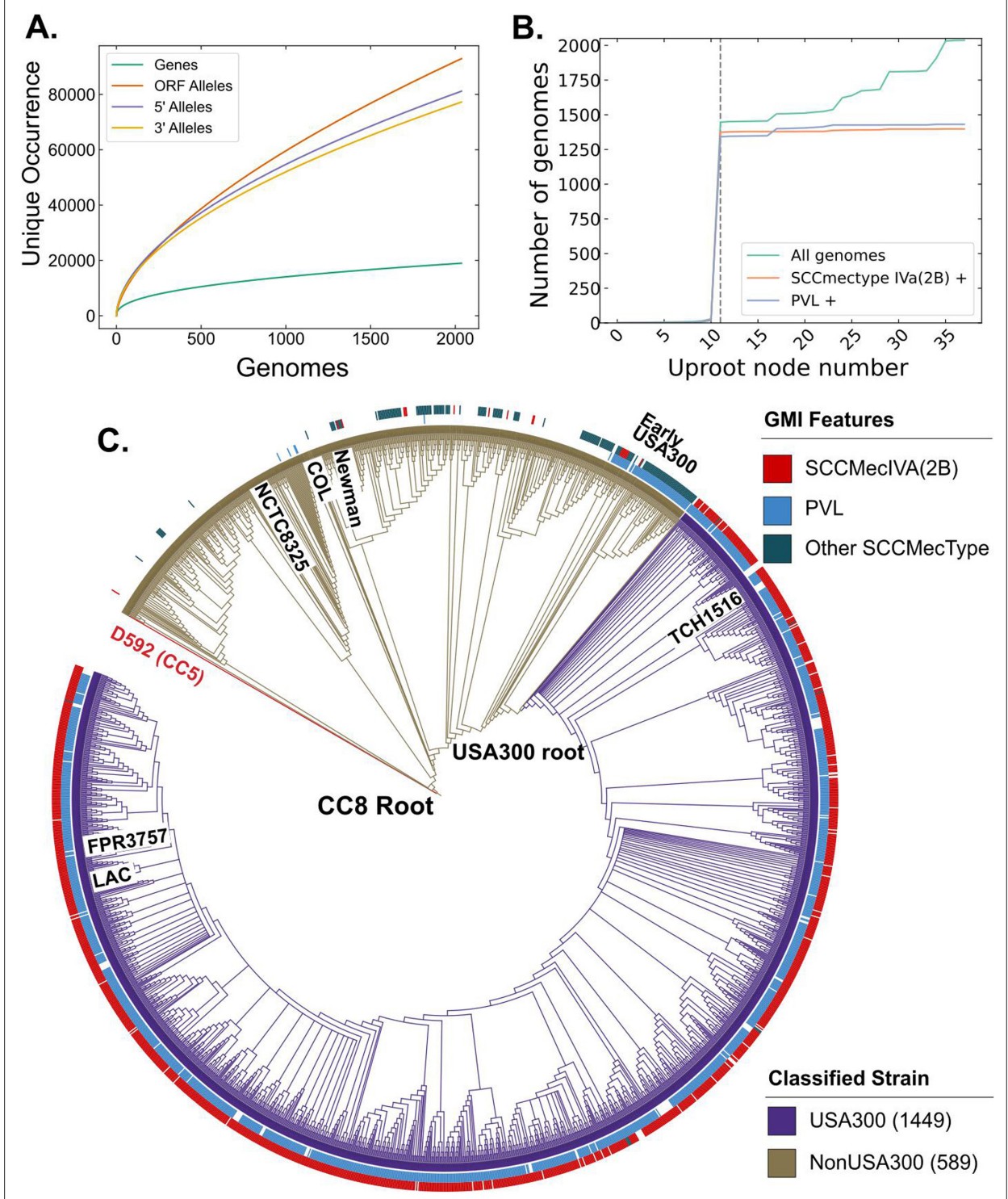

**Figure 1.** Clonal complex 8 (CC8) pangenome and phylogeny. (**A**) Pangenomic analysis of CC8 genomes shows the distribution of genes and mutations in open reading frames (ORFs) and regulatory regions. (**B**) Prevalence of USA300-specific genetic markers, Panton–Valentine Leukocidin (PVL) and SCC*mec* IVa, as you traverse up the phylogenetic tree from TCH1516. The gray dashed line represents the node where the USA300 root is placed. (**C**) Phylogenetic tree of CC8 genomes classified into USA300 and non-USA300 strains.

The online version of this article includes the following figure supplement(s) for figure 1:

**Figure supplement 1.** Pangenome analysis and strain classification.

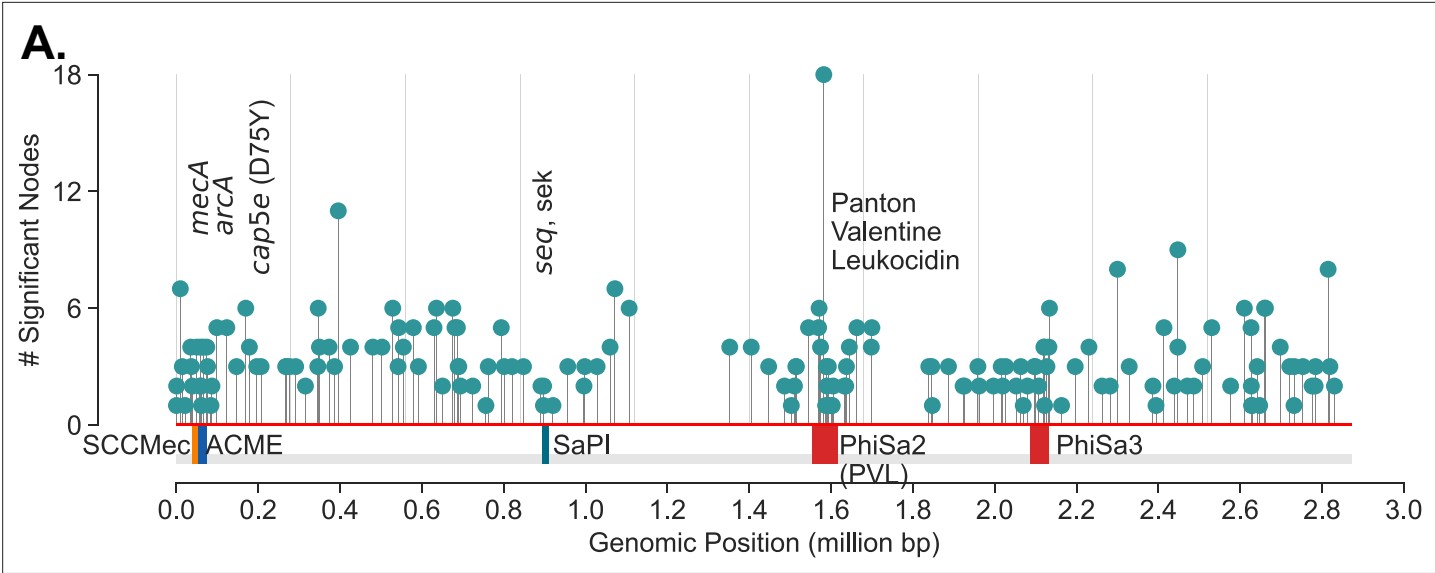

**Figure 2.** USA300 strains associated mutations. (**A**) De Bruijn graph genome-wide association study (DBGWAS) recovers components associated with USA300 previously described markers of USA300 strains including *mecA* (SCC*mec* IVa), *arcA* (ACME), *cap5e* mutation, *seq, sek*, and Phi-PVL. In addition, components with many other mutations scattered throughout the genome (NC_010079) are also enriched. Each 'significant node' represents a k-mer sequence (with minimum size of 31 nucleotides) that are associated with USA300 strains (adjusted p-value <0.05).

The online version of this article includes the following figure supplement(s) for figure 2:

**Figure supplement 1.** *S. aureus* multilocus sequence type (MLST) distribution of genomes from PATRIC used in this study.

**Figure supplement 2.** Interpreting De Bruijn graph genome-wide association study (DBGWAS) output.

k-mers that fall within the same component. We took the output component graphs and automatically extracted the enriched genetic changes, for example indels, single-nucleotide polymorphisms (SNPs), phage insertions, etc.

Many of the components were associated with genes and genetic elements expected to be enriched with USA300 strains – SCC*mec* IVA (the GMI marker), Arginine Catabolite Repressor Element (ACME), *cap5E* point mutation, multiple prophages, etc. (*Supplementary file 2*). In total, we found k-mers in 149 components associated with 137 unique TCH1516 genes that were enriched in this analysis, pointing to a large array of mutations that are unique to the USA300 lineage (*Figure 2A*). Significant k-mers in some components did not uniquely match to TCH1516 genes or only matched to genes in non-USA300 reference genome, NCTC8325.

## Genome-wide linkage and de novo mutations obfuscate identification of causal mutations

Though these mutations were enriched in USA300 strains with DBGWAS, we could not attribute the prevalence of any mutation to selection due to strong genome-wide linkage. We quantified the linkage disequilibrium by calculating the square of the correlation coefficient ($r^2$) for each of the enriched k-mer not associated with mobile genetic elements. High correlation coefficient indicates tight co-occurrence of k-mers in the genomes and therefore high linkage disequilibrium between the sequences. There was a strong linkage between the k-mers that were enriched in USA300 strains. Surprisingly, even k-mers that were 1.4 million base pairs away (the maximum distance between two sites in the circular 2.8 million base pairs long *S. aureus* genome) still had $r^2$ over 0.9 (*Figure 3A*).

To differentiate potential causal mutations from genetically linked alleles, we searched for mutation hotspots by comparing the positions of USA300 mutations in ORFs to mutations in other clonal complexes. Barring recombination events, presence of mutation hotspots in the same position in multiple clades could point to selection acting on the sequence. Therefore, we searched for prevalence of enriched mutations in other non-CC8 clades. We identified 61 SNPs within ORFs that were enriched in USA300 strains. To identify mutational hotspots in other clades, we downloaded all the amino acid sequences belonging to the PATRIC genus protein family of each of the gene products encoded by

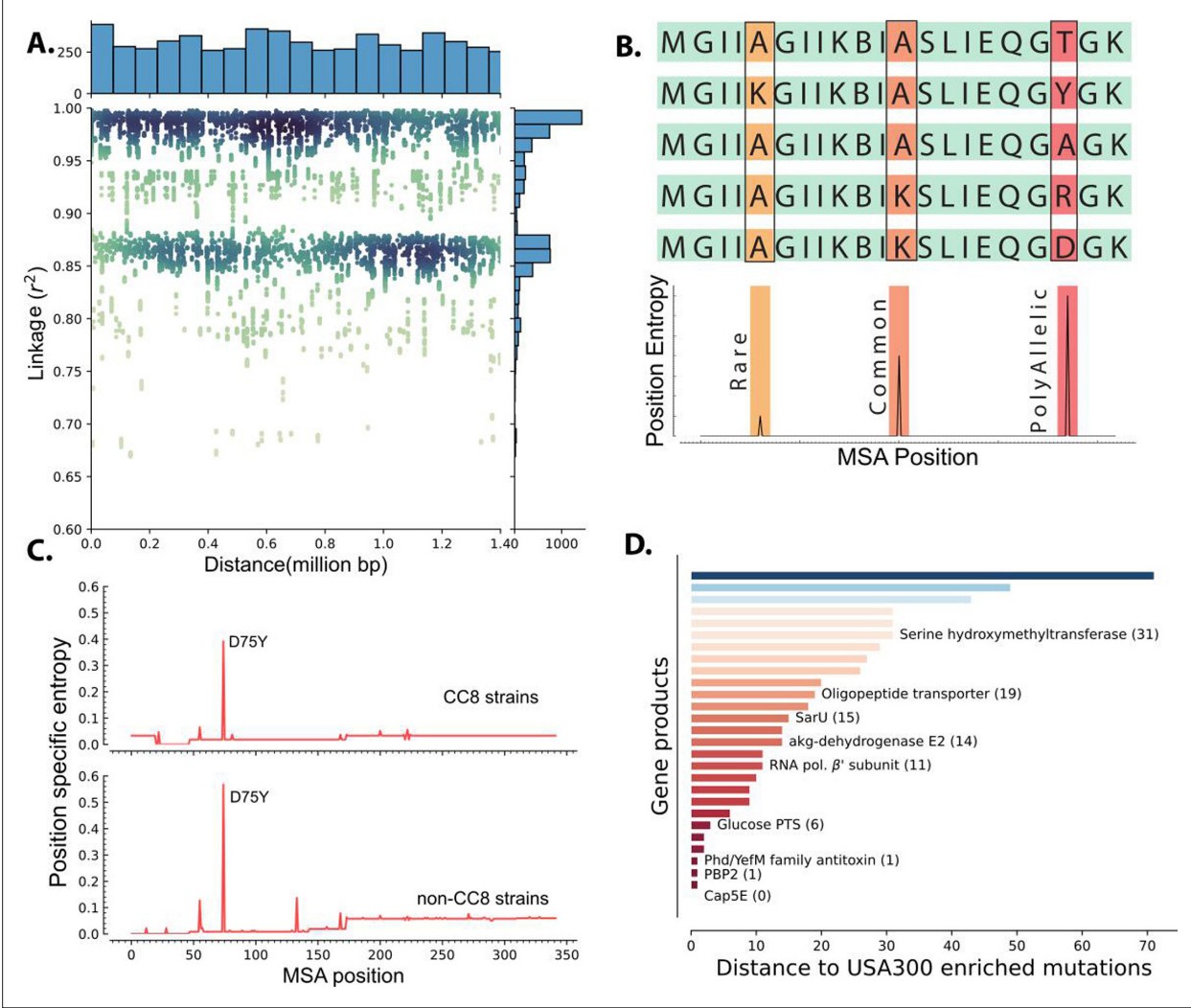

**Figure 3.** Linkage disequilibrium and de novo mutations in USA300 strains. (**A**) Enriched k-mers showed high linkage disequilibrium, with some k-mers at 1.4 Mbp distance still having $r^2$ of greater than 0.98. (**B**) Schematic of position-specific entropy analysis. Positions with heterogeneous sequences have higher calculated entropy than more conserved sequences with fewer mutations. (**C**) Using position-specific entropy, we only found one example of shared enriched mutation in open reading frames (ORFs) of USA300 and non-USA300 strains. (**D**) Distance (in base pairs) between the position of enriched mutation in USA300 strains and the position of the nearest entropy peak in other non-clonal complex 8 (CC8) strains.

The online version of this article includes the following figure supplement(s) for figure 3:

**Figure supplement 1.** SCC*mec*/ACME iModulons weighting and strain-specific activity.

the selected ORFs (*Wattam et al., 2014*). The PATRIC local protein family consists of sequences of homologous proteins within the same genus which were further filtered down to *S. aureus* species-specific sequences. After filtering, each protein family comprised 2000–16,000 unique sequences and the strains from which the amino acid sequences were derived spanned dozens of clades allowing for broad comparisons (*Figure 2—figure supplement 1*). Lastly, we removed sequences associated with ST239 as it is thought to have emerged from large-scale recombination of ST8 and ST30 strains (*Robinson and Enright, 2004*).

We determined mutation hotspots by calculating position-specific allelic entropy. Allelic entropy at a given amino acid position is a function of the number of unique amino acids found in that position and the frequency of the mutation (*Hyun et al., 2022*). Positions where all queried sequences have the same amino acid have low entropy, while positions that have frequent amino acid substitutions (hotspots) have high entropy (*Figure 3B*). This measure allows us to quickly determine the positions of mutation hotspots while accounting for multiple possible amino acid substitutions and rare

mutations. Before calculating the position-specific entropy, all sequences within each of the PATRIC local protein families were aligned with multiple sequence alignment (MSA). This alignment ensures proper comparison of amino acids even when there are deletions or insertions in some of the genes in the family.

Of the 36 enriched ORF mutations only the Asp75Tyr mutation in the *cap5E* gene, which was previously shown to ablate capsule production in USA300 strains, was found in other strains (*Figure 3C*; *Boyle-Vavra et al., 2015*). Peaks in entropy corresponding to this mutation position were present in both the CC8 and non-CC8 strains while all other mutation positions were unique to CC8. Despite not having any perfect matches outside of the *cap5E* mutation, we found that for 28 of the mutations, a peak was present in sequences from other clades within 71 MSA positions. Together, our data suggest that mutations within ORFs in USA300 strains are likely de novo mutations and are not acquired through horizontal gene transfer though many of these mutations have occurred in hotspot regions (*Figure 3D*).

## iModulon in the CC8 TRN points to mutations associated with differential regulation

The presence of genome-wide linkage and de novo mutations in ORFs severely limited the ability to distinguish causal SNPs contributing to increased pathogenesis in USA300 strains. The effect of some mutations, especially in ORFs, has been successfully linked to distinct phenotypes such as the absence of a capsule in USA300 and USA500 strains (*Boyle-Vavra et al., 2015*). However, the effect of

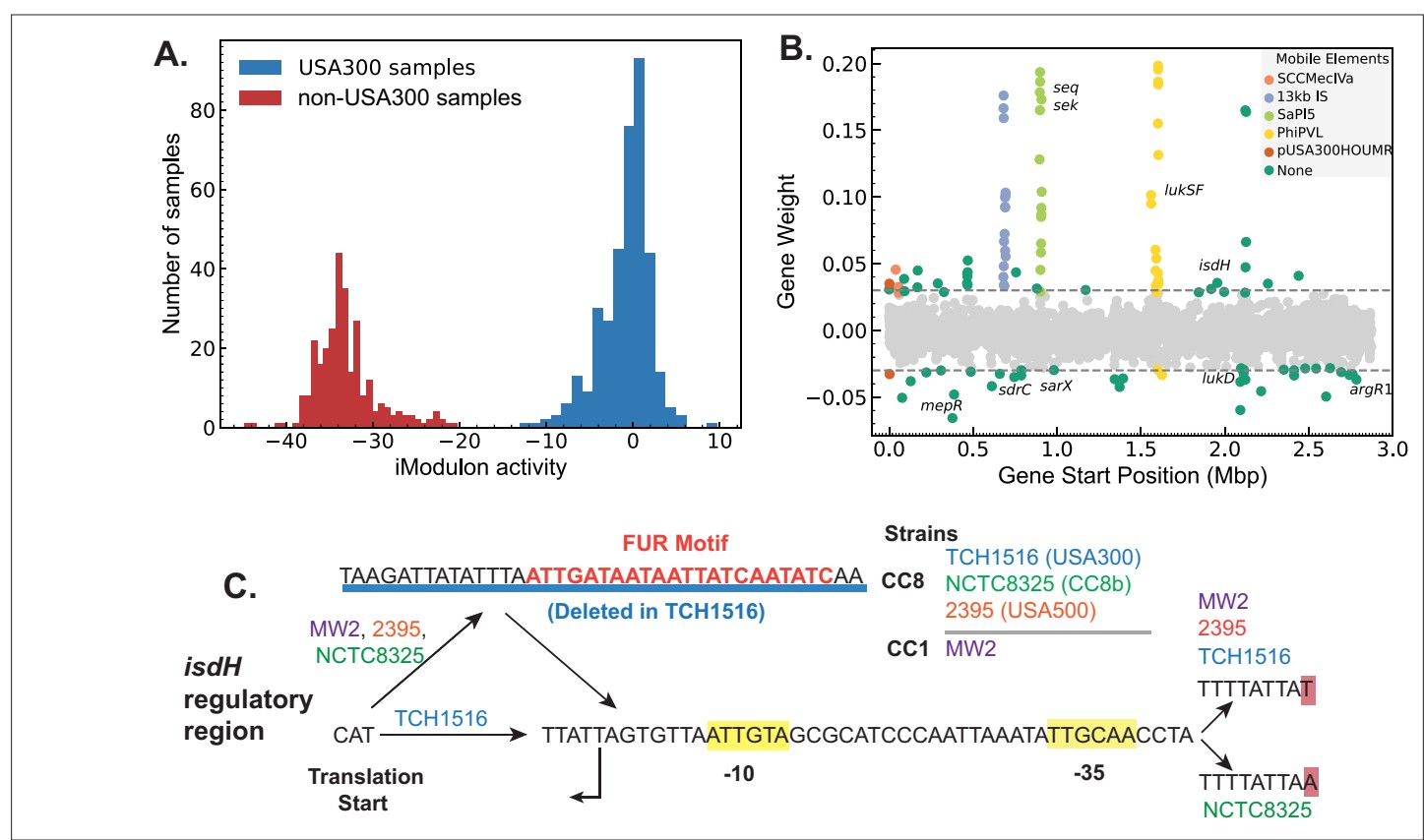

**Figure 4.** Strain-specific regulatory changes in the clonal complex 8 (CC8) clade. (**A**) Independent component analysis (ICA) of USA300 and non-USA300 RNA-sequencing data identified an iModulon with strain-specific activity. (**B**) The strain-specific iModulon contained various horizontally acquired elements (e.g. ACME, Phi-PVL) that are prevalent in USA300 lineage as well as conserved genes with strain-specific expression patterns. (**C**) Comparing the 5' regulatory region of the gene *isdH* from various *S. aureus* strains revealed a unique deletion containing Fur-binding site in USA300 reference strain TCH1516.

The online version of this article includes the following figure supplement(s) for figure 4:

**Figure supplement 1.** *isdH* gene shows strain-specific gene expression level.

mutations associated with changes in gene regulations can be much more difficult to assess (*Thurlow et al., 2012*). To look for mutations that may be associated with changes in transcriptional regulation, we used ICA to model gene regulation in CC8 strains which can predict strain-specific differences in expression patterns.

We collected CC8 strains associated RNA-sequencing data from Sequence Read Archive (SRA). After stringent QC/QA and curation, 291 non-USA300 strains (e.g. Newman, NCTC8325) and 379 USA300 (e.g. LAC, TCH1516) samples were used to create a single reconstruction of TRN across CC8 using ICA (*Sastry et al., 2019*; *Sastry et al., 2021*; *Supplementary file 3*). ICA calculates independently modulated sets of genes, iModulons, and the activities of those gene sets in each sample. iModulons calculated by ICA represent distinct sources of signals in the RNA-sequencing data. While most of the signals can be associated with different regulatory elements, iModulons associated with other biological features such as mobile genetic elements, genomic backgrounds are also enriched. In *Escherichia coli* and *Salmonella enterica Typhimurium*, multi-strain ICA has been used to calculate strain-specific iModulons that represent differences in gene expression (*Sastry et al., 2019*; *Yuan et al., 2022*).

In our reconstruction, two iModulons captured a large number of genes with different expression levels in the non-USA300 and USA300 strains (*Figure 4A*, *Figure 3—figure supplement 1A*). Most of the genes in the strain-specific iModulons belonged to mobile elements associated with USA300 strains such as ACME, SCC*mec*, Phi-PVL, etc. However, the iModulons also contained core genes that are present in both strains, pointing to possible differences in gene regulation (*Figure 4B*, *Figure 3—figure supplement 1B*).

We mapped the enriched mutations from DBGWAS onto the core genes enriched in the strain-specific iModulon. 10 core genes – including *isdH*, *argR1*, and *araC* family regulator – with mutations in the ORF or in the regulatory region were also enriched in the strain-specific iModulon (*Supplementary file 4*). Of these genes, gene *isdH*, encoding a heme scavenger molecule showed distinct strain-specific expression levels and had enriched k-mers that are mapped to the upstream regulatory region. Therefore, we compared the upstream regulatory region of several reference strains including TCH1516 (USA300), NCTC8325 (CC8b), and 2395 (USA500). Additionally, we included MW2 (CC1 CA-MRSA) as the transcription start site in the region has been experimentally confirmed in this strain (*Prados et al., 2016*). Comparisons showed a 38-bp deletion in the 5′ untranslated region containing a transcription factor Fur-binding site (*q*-val = 0.033e−4) (*Figure 4C*).

This deletion was detected in all of the 1385 USA300 genomes, but only present in 95 of the 589 non-USA300 genomes. As Fur is a repressor that blocks expression in presence of iron concentration, this deletion in the Fur-binding site may be responsible for the general increase in *isdH* expression observed in USA300 samples (*Figure 4—figure supplement 1*). We also found a second mutation upstream of the predicted −35 binding site that was also enriched in USA300. Interestingly, while the MW2 strain did not have the 38 bp deletion, it contained the exact upstream A->T mutation. All other base pairs in the region were perfectly matched in between all the reference genomes. The combination of evidence from genetic and transcriptomic analysis suggests that regulation of *isdH* is altered in USA300 strains compared to its non-USA300 progenitors.

As with many point mutations detected in our analysis, the absence of Fur-binding site upstream of *isdH* gene is prevalent only in the USA300 lineage. We searched for Fur-binding motif in the 100-bp upstream regions from 3515 non-CC8 strains spanning multiple clonal complexes (*Figure 2—figure supplement 1*). We detected the binding motif in all but 21 strains. Of the 21 strains with no detectable Fur-binding sites, 6 belonged to ST72 (out of 28 total from this type) and 6 had uncharacterized multilocus sequence type (MLST). The rest were distributed among types 121, 1750, 375, 1, 3317, 15, 7, 398, and 4803, with one positive strain per type.

## Discussion

Emergence of CA-MRSA USA300 strains from hospital-associated methicillin-resistant *S. aureus* USA500 progenitors presents a natural experiment to probe the genetic basis for the establishment of the USA300 lineage. However, in studying these groups, genetic methods like GWAS were limited in finding causal mutations due to genome-wide linkage disequilibrium and presence of an unexpectedly large number of de novo mutations unique to the USA300 lineage. Here, we demonstrated how a model of transcriptional regulation with iModulons can be used to make a headway through

the impasse created by the high linkage disequilibrium and identify GWAS-enriched mutations that are also associated with measurable phenotypic changes in the TRN. From the combined RNA-sequencing dataset of USA300 and non-USA300 strains, ICA calculated two iModulons that captured strain-specific variation in gene expression. As expected, most genes in the iModulons were part of mobile genetic elements such as ACME and SCC*mec* because they have zero expression level in non-USA300 samples. However, the iModulon also contained several core genes that are present in both groups but are differentially regulated. A deeper analysis of the regulatory region of one of these genes with enriched mutation, *isdH*, revealed a deletion of a DNA segment containing the binding site of the Fur repressor. In congruence with this observation, we also found that USA300 strains with the deleted Fur-binding site showed general increase in *isdH* expression level. This gene encodes IsdH, a surface receptor that binds to human hemoglobin, causing it to release the heme (***Ellis-Guardiola et al., 2020***). It is part of an arsenal of *S. aureus* iron sequestration proteins including Staphyloferrins and ferrous iron transporters that it uses to compete with the host for essential iron (***van Dijk et al., 2022***). Despite having many different pathways for obtaining iron, it has been observed that *S. aureus* prefers heme as its iron source over transferrin-bound iron (***Skaar et al., 2004***). Our analysis shows that preference for heme is reflected in the genomic and transcriptomic signature of USA300 as a deletion of Fur-binding region upstream of *isdH* and subsequent increase in its expression. Combining GWAS with large-scale transcriptomic modeling was therefore able to predict potential causal mutations contributing to the increased clinical burden of the USA300 lineage.

The current analysis utilized the available DNA- and RNA-sequencing data and the methods used here are scalable to the rapidly growing number of data in the public repositories. Indeed, with the greater scale, we can get more granular insight into subclade-specific differences. The transcriptomic analysis consisted of samples primarily from the USA300 (CC8e and CC8f) clades, the CC8a clade represented by Newman and the CC8b clade represented by NCTC8325 and its derivatives. However, the CC8b and CC8a clades are currently undersampled due to its minimal clinical burden compared to USA300. We therefore combined strains from all non-USA300 clades into a single group for GWAS. The misalignment of RNA-sequencing samples from GWAS samples may explain the low number of hits that were enriched by both methods when many other unique gene expression patterns have been observed in USA300 strains. This misalignment points to the limit of our approach. Most other phenotypes of clinical interest such as antibiotic resistance may not separate cleanly into distinct clades. In those cases, it is not obvious which strains should be chosen as the reference strain for RNA-sequencing and subsequent TRN reconstruction. The choice of reference strain as well as the choices in the RNA-sequencing sample conditions will impact which association between mutations and changes in gene regulations are uncovered.

With time, the scaling of databases may be able to resolve the issue of imbalanced sampling. On the other hand, resolving the confounding effect of linkage disequilibrium inherent in emerging and clonal strains will require a new generation of modeling methods (***Bal et al., 2016***). Our current approach focuses on modeling the changes in gene regulation at the transcriptional level, but causal mutations can have any number of effects on the phenotype of the organism. New modeling methods that can systematically predict these other phenotypes are now rapidly emerging. Our recent work with *Mycobacterium tuberculosis* utilized a metabolic allele classifier which combines genome scale metabolic models with machine learning to estimate biochemical effects of alleles thus mapping mutations to changes in metabolic fluxes (***Kavvas et al., 2020***). Similarly, advances in protein structure prediction with AlphaFold2 and RosettaFold puts us at the cusp of predicting the effects of mutations on protein folding (***Baek et al., 2021***; ***Jumper et al., 2021***). Combination of these modeling techniques may therefore prove to be the breakthrough required to advance solutions to the current challenges in population genetics of emerging pathogens.

## Materials and methods
### Pangenomic analysis
The pangenome analysis was run as described in detail before (***Hyun et al., 2022***). Briefly, 'complete' or 'WGS' samples from CC8/ST8 were downloaded from the PATRIC database (***Wattam et al., 2014***). Sequences with lengths that were not within 3 standard deviations of the mean length or those with more than 100 contigs were filtered out. A non-redundant list of CDSs from all genomes was created

and clustered by protein sequence using CD-HIT (v4.6) with minimum identity (-T) and minimum alignment length (-aL) of 80% and word size of 5 (-w 5) (*Fu et al., 2012*). To get the 5′ and 3′ sequences, non-redundant 300 nucleotide upstream and downstream sequences from the CDS were extracted for each gene.

The CDSs were divided into core, accessory, and unique genes based on the frequency of genes as previously described (*Hyun et al., 2022*). To calculate the frequency thresholds for each category, $P(x)$, the number of genes with frequency $x$ and its integral $F(x)$, the cumulative frequency less than or equal to $x$ were calculated. The multimodal gene distribution can be estimated by sum of two power laws as:

$$P\left(x\right) = c_1 x^{-\alpha_1} + c_2 \left(N + 1 - x\right)^{-\alpha_2} x = 1, 2, ..., N$$

where $N$ is the total number of genomes, $x$ is the gene frequency and ($c_1$, $c_2$, $-\alpha_1$, $-\alpha_2$) are parameters fit based on the data. The cumulative distribution is then the integral of $P(x)$ with additional parameter $k$:

$$F\left(x\right) = k + \frac{C_1}{1 - \alpha_1} x^{1-\alpha_1} \frac{-C_2}{1 - \alpha_2} \left(N + 1 - x\right)^{1-\alpha_2}$$

The parameters ($c_1$, $c_2$, $-\alpha_1$, $-\alpha_2$, and $k$) were fitted based on the data using nonlinear least squares regression from scipy (*Jones et al., 2001*). The frequency threshold of core genomes was defined as greater than $0.9N + 0.1x^*$ and the threshold for unique genome was defined as $0.1x^*$, where $x^*$ represents the inflection point of the fitted cumulative distribution.

Roary (v3.13.0) was used with -i 95 flag to confirm the output of our pangenome analysis (*Page et al., 2015*).

## Reconstructing CC8 phylogenetic tree

The phylogenetic tree was reconstructed using the standardized PHaME pipeline on the PATRIC sequences that passed the QC/QA (*Shakya et al., 2020*). Using the pipeline, the contigs and sequences were aligned to the reference TCH1516 genome NC_010079 and plasmids NC_012417, NC_010063 (*Highlander et al., 2007*) and 24881 core SNPs at were calculated. The core SNPs were then used to estimate the phylogenetic tree using IQ-TREE (v1.6.7) run with 1000 bootstraps and utilizing the ultrafast bootstrap (*Minh et al., 2020*; *Hoang et al., 2018*). The tree was built using the 'TVMe+ASC+G4' model as suggested by the IQ-TREE ModelFinder (*Kalyaanamoorthy et al., 2017*). Finally, iTOL was used to visualize, annotate, and root the tree with the USA100 D592 (NZ_CP035791) from CC5 as the outgroup (*Letunic and Bork, 2021*).

## Classification of USA300 and non-USA300 strains

The USA300 and non-USA300 strains were classified based on a previously proposed and validated CC8 subtyping scheme (*Bowers et al., 2018*). In this scheme, USA300 strains can be identified from the whole genome if they are PVL positive MSSA or MRSA with SCC*mec* IVa cassette. We detected SCC*mec* types using SCC*mec*Finder (v1.2), and only those genomes where the cassette could be identified by both BLASTn and k-mer based methods were marked as positive (*Kaya et al., 2018*). PVL was detected using nucleotide BLAST (v2.2.31). We added additional criteria that all genomes identified as USA300 by GMI form a distinct subclade before they are labeled as USA300 that is PVL or SCC*mec* IVa positive genomes that grouped separately from other USA300 strains in the phylogenetic tree were not labeled as USA300. To find the root of the USA300 strains in the phylogenetic tree, the genomes in the tree were first annotated by their PVL and SCC*mec* status. Then the tree traversed from leaf to root starting from known USA300 strains – TCH1516 and FPR3757 – while keeping track of the number of descendant genomes from the current root that contained known markers SCC*mec* IVa and PVL. The node where the number of genomes with the markers started flatlining was marked as the root of USA300.

We detected SCC*mec* cassettes in 1588 genomes of which 1358 were SCC*mec* IVa positive. We also found 1431 PVL positive genomes using BLASTn search with PVL encoding genes from USA300 TCH1516 (USA300HOU_RS07645, USA300HOU_RS07650) as reference (*Supplementary file 5*). Lastly, we reconstructed the CC8 phylogenetic tree based on core SNPs and rooted the tree using strain D592 (CC5) as an outgroup. The tree was then traversed from reference strain TCH1516 to the

CC8 root using ete3, while tracking the total number of genomes, the total number of SCC*mec* IVa positive genomes and the number of PVL positive genomes in each root (*Huerta-Cepas et al., 2016*). The root of USA300 was placed manually where the number of total genomes kept increasing while the number of PVL and SCC*mec* positive genomes plateaued. All strains in the clade represented by the USA300 root were classified as USA300 regardless of their SCC*mec* or PVL status.

## DBGWAS and k-mer linkage calculations

DBGWAS (v0.5.4) was used to enrich mutations unique to USA300 strains using default k-mer size of 31 (-k 31) and neighborhood size of 5 (-nh 5). Alleles with frequency less than 0.1 were filtered (-maf 0.1) and all components enriched with *q*-values less than 0.05 were documented (-SFF q0.05). Genome-wide linkage was estimated by Pearson correlation (calculated with built-in Pandas function) of the presence/absence of enriched k-mers and distance was measured based on the k-mer alignment to the reference TCH1516 genome as determined by BLASTn.

To determine the enriched 'genetic event' (e.g. SNP, indel, mobile genetic element, etc.), the graph output from DBGWAS was first loaded onto a networkX model (*Hagberg et al., 2008*). All nodes in the graph with frequency lower than 0.05 were discarded. MGEs were identified if all significant nodes from DBGWAS had higher frequencies in one strain, for example all nodes associated with SCC*mec* had higher frequencies in USA300 strains. To find SNPs and smaller indel events, the networkx was used to find cycles in the graph, which results from bifurcation and eventual re-collapse of Debruijn graphs around mutations . For each cycle, the 'end nodes' representing the start and end of the bifurcation were identified by finding the nodes in the cycle with highest frequency across all samples (*Figure 2—figure supplement 2*). As 'end nodes' are present in both case and control samples, they will have higher frequency than other nodes in the cycle which are specific to either case or control. Once the end nodes are identified, the two paths around the bifurcations representing the case and control specific sequences were identified using the shortest path algorithm in networkx. The sequences from nodes of each path were concatenated, changing the sequences to reverse complements and removing overlaps in sequences when required. The concatenated sequences from each path were then compared using BioPython (v1.83) pairwise global alignments to find the SNPs or indels that differentiate the sequences from case and control (*Cock et al., 2009*). If reference sequences are passed, the concatenated sequences are aligned to the reference sequences using nucleotide BLAST and mutation positions were converted from k-mer positions to positions in the reference genomes. The code used for this analysis can be found in https://github.com/SBRG/dbgwas-network (copy archived at *Systems Biology Research Group, 2022*).

## Mapping mutation hotspots with position-specific Shannon entropy

For each of the CDS with enriched mutations, the PATRIC local protein family (PLfam) was identified based on the reference TCH1516 genome. All available protein sequences for each CDS PLfam were downloaded and filtered for *S. aureus* sequences. The MLST of the source genome of each downloaded sequence was mapped using the PATRIC database. The online PATRIC website was used to find and filter the target sequences. The sequences were divided into ST8 and non-ST8 and ST239 sequences were filtered. MAFFT was used for multiple alignment and position-specific Shannon entropy was calculated on the aligned file (*Katoh et al., 2002*). The entropy is calculated as:

$$H\left(X\right) = -\sum_{i=1}^{n} P\left(x_i\right)\log_2 P\left(x_i\right)$$

where *n* is the total number of unique amino acids in the position and $P(x_i)$ is the probability of finding the given amino acid.

## Calculating strain-specific iModulons with ICA

ICA of RNA-sequencing data was performed using the pymodulon package (*Sastry et al., 2021*). Using the package, all available RNA-sequencing data for non-USA300 and USA300 strains were downloaded, run through the QC/QA pipeline, manually curated for metadata and aligned to the TCH1516 genome (NC_010079, NC_012417, and NC_010063). The combined data were then transformed into log-TPM (transcripts per million) and normalized to a single reference condition (SRX3760886 and SRX3760891). This contrasts with other ICA models that normalize the data to

project-specific reference conditions to reduce batch effects. However, normalizing to project-specific control conditions also erases the strain-specific information as almost all BioProjects contain data from only one isolate (e.g. NCTC8325, TCH1516, LAC, etc). ICA was then run following our previously established pipeline to generate iModulons for CC8 clade *S. aureus* (*Sastry et al., 2019*). This process is described in further detail below.

We began by collecting all available RNA-sequencing data and metadata for *S. aureus* strains from SRA that belonged to the CC8 clade. Most of the sequences were from well-studied CC8 strains – TCH1516, FPR3757, LAC, Newman, and NCTC8325. Others had fewer specific labels for example 'USA300' but still belonged to CC8. The fastq files for the samples were trimmed with TrimGalore (v0.6.5) and aligned to reference TCH1516 genome (NC_010079, NC_012417, and NC_010063) with bowtie2 (v1.2.3) (*Krueger, 2015*; *Langmead and Salzberg, 2012*). The gene-specific read counts were calculated with HTSeqCount (v2.0.1) using the intersection-strict criteria. The number of mapped reads was then normalized to TPM and log-transformed (log-TPM).

Before using the data, the quality of the reads and alignment was assessed using FastQC and MultiQC (v 1.11) (*Andrews, 2010*; *Ewels et al., 2016*). Any samples failing 'per base sequence quality', 'per sequence quality score', 'per base *n* content', or 'adapter content' were dropped. Additionally, we also removed samples with less than 500,000 reads aligned to the reference. Lastly, samples that did not contain replicates or those with replicates with Pearson correlation coefficients less than 0.9 were also excluded. We then collected additional metadata for the remaining 670 RNA-sequencing samples including growth conditions, genetic changes, associated experiment, etc. The log-TPM were then centered to the reference condition of *S. aureus* TCH1516 grown in RPMI+10% LB. By centering data from non-USA300 strains on USA300 reference, ICA is able to pick up strain-specific regulatory changes for example ICA captures the activity of Fur transcription factor as a linear combination of Fur iModulon containing gene regulated by Fur and a second 'strain-specific' iModulon that captures differences between USA300 and non-USA300 strains (*Figure 3—figure supplement 1*).

We applied FastICA to the centered log-TPM to calculate the M and the A matrix which, respectively, describe the iModulon structure and their activities (*Pedregosa, 2011*; *Koldovský et al., 2006*). To find the best possible model, we first had to compute an optimal number of stable components. As FastICA is non-deterministic, each iteration yields a slightly different component weightings and activity levels. It may also yield 'spurious' components that are only present in a subset of runs. To find stable components, we ran ICA 100 times with a random seed. Similar components (e.g. same component containing Fur-associated iModulon) from different iterations which may have slightly different weightings were detected by clustering with DBSCAN. Only components that appear in every run were accepted. When running ICA, users must also provide the number of desired components that the data will be decomposed into. Decomposition into too few components could lead to signals from several transcription factors being combined into single components while over decomposition leads to many unstable and 'single-gene' iModulons that likely capture noise in the data. To find the optimal number of components we used a heuristic method, OptICA, which runs ICA with different numbers of input components from 10 to 340 and suggests an optimal component that minimizes single-gene iModulons while maximizing robust components (*McConn et al., 2021*). Based on this heuristic, the final model was built with 270 components as input, 148 of which were determined to be robust components.

In each component, we labeled a gene as being part of an iModulon if their weighting in that component did not fall within a Gaussian distribution as determined by D'Agostino's test. The genes in each iModulon were then compared to genomic features (e.g. regulons, phage, mobile cassettes, etc.; see 'TRN' object in the model), and was determined to be associated with the feature if there was significant overlap between the two groups (hypergeometric test; adjusted p-value <0.05, precision ≥0.5, and coverage ≥0.2). We also manually curated other iModulons associated with other features for example iModulon where all member genes associated with translation were labeled 'Translation iModulon'. The activities of the output iModulons were manually parsed to look for iModulons with the largest strain-specific differences.

## Fur box motif search

*isdH* genes in all the genomes were first clustered using CD-HIT with identity and coverage minimum of 0.8 (*Fu et al., 2012*). All annotated *isdH* genes fell within a single cluster. For each genome, the

100 bp upstream region was then extracted and used for motif search. Motif search for the Fur box was conducted using the FIMO package from the MEME suite (v5.1.0) with default settings (*Bailey et al., 2009*). The *S. aureus* strain NCTC8325 Fur motif from collecTF was used as a reference (*Kiliç et al., 2014*).

## Additional information

### Funding

| Funder | Grant reference number | Author |
|---|---|---|
| National Institute of Allergy and Infectious Diseases | AI124316 | Saugat Poudel<br>Jason Hyun<br>Ying Hefner<br>Jon Monk<br>Victor Nizet<br>Bernhard O Palsson |

The funders had no role in study design, data collection, and interpretation, or the decision to submit the work for publication.

### Author contributions

Saugat Poudel, Conceptualization, Data curation, Formal analysis, Validation, Investigation, Visualization, Methodology, Writing – original draft, Writing – review and editing; Jason Hyun, Formal analysis, Investigation, Methodology, Writing – original draft, Writing – review and editing; Ying Hefner, Investigation, Methodology; Jon Monk, Investigation, Methodology, Writing – original draft, Writing – review and editing; Victor Nizet, Conceptualization, Supervision, Funding acquisition, Writing – original draft, Project administration, Writing – review and editing; Bernhard O Palsson, Conceptualization, Resources, Supervision, Funding acquisition, Validation, Investigation, Project administration, Writing – review and editing

### Author ORCIDs

Saugat Poudel ⓘ https://orcid.org/0000-0002-3732-2463
Victor Nizet ⓘ https://orcid.org/0000-0003-3847-0422
Bernhard O Palsson ⓘ https://orcid.org/0000-0003-2357-6785

Reviewer #1 (Public Review): https://doi.org/10.7554/eLife.90668.3.sa1
Author response https://doi.org/10.7554/eLife.90668.3.sa2

## Additional files

### Supplementary files

Supplementary file 1. List of *S. aureus* strains used for De Bruijn graph genome-wide association study (GWAS) analysis.

Supplementary file 2. Raw De Bruijn graph genome-wide association study (DBGWAS) output and network analysis output.

Supplementary file 3. Metadata of the RNA-sequencing data used to create the transcriptional regulatory network (TRN) model with independent component analysis (ICA).

Supplementary file 4. Conserved genes that were enriched by De Bruijn graph genome-wide association study (DBGWAS) and by iModulon analysis.

Supplementary file 5. Results of BLASTing Panton–Valentine Leukocidin (PVL) genes against all genomes used in De Bruijn graph genome-wide association study (DBGWAS).

MDAR checklist

### Data availability

All RNA-sequencing data used to create the model of the TRN are available in SRA (see *Supplementary file 3* for accession numbers). All genomes used in DBGWAS can be found in the PATRIC

database (see *Supplementary file 1* for details). The code used for analysis, the intermediate files and models are available on GitHub (https://github.com/sapoudel/USA300GWASPUB; copy archived at *Poudel, 2024*).

The following datasets were generated:

| Author(s) | Year | Dataset title | Dataset URL | Database and Identifier |
|---|---|---|---|---|
| Poudel S, Szubin R, Palsson BO | 2024 | Effect of different carbon sources on gene expression of *S. aureus* USA300 JE2 | https://www.ncbi.nlm.nih.gov/bioproject/?term=PRJNA1086018 | NCBI BioProject, PRJNA1086018 |
| Poudel S, Szubin R, Palsson BO | 2022 | *Staphylococcus aureus* JE2 TF knockout gene expression | https://www.ncbi.nlm.nih.gov/bioproject/?term=PRJNA881834 | NCBI BioProject, PRJNA881834 |
| Poudel S, Szubin R, Palsson BO | 2022 | Metabolic perturbation of *S. aureus* USA300 JE2 | https://www.ncbi.nlm.nih.gov/bioproject/?term=PRJNA872284 | NCBI BioProject, PRJNA872284 |

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
