## [Editor Report · eLife assessment]

This study presents **valuable** findings on core genome mutations that might have driven the emergence of the *Staphylococcus aureus* lineage USA300, a frequent cause of community-acquired infections. The authors present a **solid** novel approach that combines genome-wide association studies and RNA-expression analyses, both applied to extensive publicly available datasets. This approach generated an intriguing hypothesis that should be validated experimentally. The work will interest microbiologists working in genomic epidemiology and phenotype-genotype association studies.

---

## [Referee Report · Reviewer #1 (Public Review)]

Summary:

This is large-scale genomics and transcriptomics study of the epidemic community-acquired methicillin-resistant *S. aureus* clone USA300, designed to identify core genome mutations that drove the emergence of the clone. It used publicly available datasets and a combination of genome-wide association studies (GWAS) and independent principal-component analysis (ICA) of RNA-seq profiles to compare USA300 versus non-USA300 within clonal complex 8. By overlapping the analyses the authors identified a 38bp deletion upstream of the iron-scavenging surface-protein gene isdH that was both significantly associated with the USA300 lineage and with a decreased transcription of the gene.

Strengths:

Several genomic studies have investigated genomic factors driving the emergence of successful *S. aureus* clones, in particular USA300. These studies have often focussed on acquisition of key accessory genes or have focussed on a small number of strains. This study makes a smart use of publicly available repositories to leverage the sample size of the analysis and identify new genomics markers of USA300 success.

The approach of combining large-scale genomics and transcriptomics analysis is powerful, as it allows to make some inferences on the impact of the mutations. This is particular important for mutations in intergenic regions, whose functional impact is often uncertain.

The statistical genomics approaches are elegant and state-of-the-art and can be easily applied to other contexts or pathogens.

Weaknesses:

The main weakness of this work is that these data don't allow a casual inference on the role of isdH in driving the emergence of USA300. It is of course impossible to prove which mutation or gene drove the success of the clone, however, experimental data would have strengthen the conclusions of the authors in my opinion.

Another limitation of this approach is that the approach taken here doesn't allow to make any conclusions on the adaptive role of the isdH mutation. In other words, it is still possible that the mutation is just a marker of USA300 success, due to other factors such as PVL, ACMI or the SCCmecIVa. This is because by its nature this analysis is heavy influenced by population structure. Usually, GWAS is applied to find genetic loci that are associated with a phenotype and are independent of the underlying population structure. Here, authors are using GWAS to find loci that are associated with a lineage. In other words, they are simply running a univariate analysis (likely a logistic regression) between genetic loci and the lineage without any correction for population structure, since population structure is the outcome. Therefore, this approach can't be applied to most phenotype-genotype studies where correction for population structure is critical.

Finally, the approach used is complex and not easily reproduced to another dataset. Although I like DBGWAS and find the network analysis elegant, I would be interested in seeing how a simpler GWAS tool like Pyseer would perform.

---

## [Author Response]

The following is the authors’ response to the original reviews.

**Reviewer #1 (Recommendations For The Authors):**
(1) Line 56: replace "pyomastitis" with "pyogenic skin infections".

Corrected.

(2) Line 58: replace "basal strains" with "ancestral strains".

Corrected.

(3) Line 62: population structure impacts gene acquisition too, however, gene acquisitions can be easier to connect with a phenotype. For example, acquisition of mecA is thought to be adaptive rather than just linked to a successful lineage. This same reasoning applies to resistance-associated mutations such as gyrA mutations in ST22 emergence.

We completely agree with the reviewer that population structure also impacts gene acquisition. We wanted to convey that connecting gain or loss of genes to a change in particular phenotype is much easier than doing the same for a mutation, specially in the presence of strong linkage, and therefore gene level analysis is the focus of many previous studies. We have rewritten the sentence to better convey this idea:

“Due to this limitation, studies of emerging strains often focus on gene level analysis such as acquisition of mobile genetic elements or loss of gene function as their effect on phenotype is easier to determine than that of point mutations.”

(4) Line 112 this might be simply due to the smaller size of the intergenic regions chosen. I suggest to correct for the size of the genome segment considered.

We thank the reviewer for pointing this out. The size of the intergenic was indeed the simple explanation for this observation. We have added the following sentence to the manuscript:

“This is reflective of the fact that most of *S. aureus* genome sequence comprises of ORFs e.g. ~84% of TCH1516 genome is part of an ORF.”

(5) Line 189: please add p values to supp table 2.

We have added the p and q values from DBGWAS into Supp table 2. It is under the ‘DBGWAS Result’ sheet.

(6) Line 227: high entropy indicates that this site is polymorph, not necessarily that there is selective pressure. In the extreme, this might actually point to a neutral position, since any amino-acid could be equally present (see for example https://www.nature.com/articles/s41467-022-31643-3#Sec10).

We agree that high entropy by itself may point to a position with neutral selection leading to some false positives. However, we were focused on positions that were mostly biallelic in CC8, and with differential prevalence in USA300 vs non-USA300 (albeit in the presence of strong linkage disequilibrium) in addition to having high entropy in non-CC8 strains. This helps us filter some of the positions that were mostly monoallelic or with rare mutations while preserving other sites of interest. The approach was able to find *cap5E* mutation which has been associated with disruption of capsule production.

(7) Line 271: show USA500 on the tree.

Our current study is mostly focused on differences between USA300 and non-USA300 strains and we want to highlight those differences in the tree.

(8) Line 327: still not possible to infer causality.

We have changed the language to remove mentions of causality and instead talk about the association of GWAS enriched genes with measured transcriptional changes. The revised sentence now reads:

“Here, we demonstrated how a model of transcriptional regulation with iModulons can be used to make a headway through the impasse created by the high linkage disequilibrium and identify GWAS-enriched mutations that are also associated with measurable phenotypic changes in the TRN.”

(9) Line 324: subclades reference.

We are unsure what this means.

(10) Line 366: the authors seem to have used a bespoke pan-genome analysis approach. Would they be able to validate it using established tools such as Roary, Pirate or Panaroo? Panaroo in particular appears to have superior accuracy thanks to its pan-genome graph approach (https://github.com/gtonkinhill/panaroo).

We have added the results of Roary to our analysis (Figure S1b). The roary results largely agree with our biggest take away from pangenomics which is that our collection of genomes have a good coverage of the CC8 clade at the gene level.

(11) Line 397: what was the size of the core genome?

There were 24881 core sites. We have added the number to the manuscript.

(12) Line 407: please add citation or website for SCCmecFinder.

The citation of SCCmecFinder (45) is at the end of the sentence.

(13) Line 421: I was not able to find the code used for this analysis in the github repository provided.

The code can be found in “notebook/02_Preprocess_DBGWAS.ipynb” within the repo.

(14) Line 427: this is a very complex analysis for a simple univariate comparison between USA300-vs-non USA300 strains with no correction for population structure. The authors should compare their results with a more established pipeline like Pyseer or Gemma that can handle kmers and show the added value of their approach.

We wanted to take advantage of DBGWAS’s ability to collapse kmers into unitigs and further collapse significant unitigs within a genetic neighborhood into components. Unfortunately, we found that in many cases, it became difficult to determine the exact mutation that was being enriched e.g. (T234G) without doing lots of manual work. Our network analysis simply parses the DBGWAS graph to automatically extract these mutations, making the results more interpretable. It does not do any additional hypothesis testing.

We also attempted to pass kmer data into GEMMA but without the compaction provided by DBGWAS the memory required (>168 GB) exceeded what we had available.

(15) DBGWAS: please indicate DBGWAS version and the options used for kmer size and number of neighbour nodes retained in the subgraph. Also, I assume that no correction for population structure was applied.

We have added the version and parameters for DBGWAS. The method section now reads:

“DBGWAS (v0.5.4) was used to enrich mutations unique to USA300 strains using default kmer size of 31 (-k 31) and neighborhood size of 5 (-nh 5). Alleles with frequency less than 0.1 were filtered (-maf 0.1) and all components enriched with q-values less than 0.05 were documented (-SFF q0.05).”

(16) Could the authors provide the DBGWAS output for the most significant unitings in graph format? This would help readers understand the findings.

The outputs are available in the github repo. The link to this specific data is (https://github.com/sapoudel/USA300GWASPUB/tree/master/data/dbgwas/dbgwas_output/visualisations)

The text format of the output is part of Supplementary Table 2 under “DBGWAS Result” sheet.

(17) Line 469: please provide more details on iModulons, it is not enough to simply reference the paper: specific QC criteria, mapping algorithm and parameters, ICA algorithm.

We have now added a new Supplementary Note 2 section with more details about building iModulons.

(18) Line 474: what is log-TPM?

Log-Transcripts per Million. We have added the description in the text.

(19) Line 479: not sure what "Chapter 3" refers to.

Thank you for correcting the mistake. The reference has been corrected.

**Reviewer #2 (Recommendations For The Authors):**
Line 45. The introduction is not well-structured, and there is a lack of coherence among the topics pertinent to the research objective. I would recommend rewriting this section addressing the following topics: the challenge of distinguishing lineages within the CC8, especially the CA-MRSA USA300 strains; discussing the state-of-the-art GWAS methodologies, elucidating the main confounding factors in the application of GWAS to bacterial studies, and finally, exploring how current methods aim to address these concerns.

We would like to thank the reviewer for the suggestions. The main innovation of the paper is using iModulons to find phenotype associated mutations from a set of linked mutations. The challenge of distinguishing CC8 subclades has been largely resolved thanks to efforts by Bowers et al. (PMID: 29720527). We have made some revisions to address the GWAS methodologies (bugwas and DBGWAS), the effect of linkage disequilibrium in interpreting the output of these methods and how combining the results of these association tests with modeling of TRN with iModulons can lead to finding candidate mutations of interest that are linked to specific changes in gene regulation.

Line 56. Replace "pyomastitis" with "pyomyositis".

Corrected to “pyogenic skin infections.”

Lines 71. What do the authors mean by "endemic USA300 strain"?

We have removed references to endemic strains.

Line 106. Please verify the number of genomes used in the DBGWAS analysis. In the text, the authors mention that 2038 genomes were utilized. However, in Supplementary Table 1, only 2030 genomes are listed.

Thank you for catching the discrepancy. We started the analysis with 2037 genomes, including four “spiked-in” reference genomes- USA100 D592 (CC5 strain used for rooting the CC8 tree), TCH1516 (same accession number as the one used for ICA), COL and Newman. Before further analysis, we removed 6 genomes for being smaller than 2.5 million base-pairs (see preprocessing.ipynb) and the USA100 D592 strain as it is not part of CC8. This resulted in 2030 genomes being used for DBGWAS. We kept the other 3 spiked CC8 genomes to help annotate the unitigs from DBGWAS. Lastly, we removed the other three CC8 clade spiked genomes for pangenomic analysis. To clarify this, we have made the following changes to the text:

(1) Changed line 106: We downloaded 2033 *S.* aureus genomes for analysis and excluded six of them with genome length of less than 2.5 million base pairs. The remaining 2027 *S. aureus* CC8 genomes formed a closed pangenome, suggesting that the sampled genomes mostly captured the gene level variations within the clonal complex (Figure 1a).

(2) DBGWAS section Line 177: We used 2030 genomes for this analysis; the 2027 genomes in pangenomics analysis above were “spiked” with three well known CC8 genomes- TCH1516, COL, and Newman- to help annotate the DBGWAS unitigs.

Line 108. Could the authors provide a table with the genes that constitute the core, accessory genome, and unique genes for each of the strains?

The genes presence absence tables are very large files and therefore we have only added them to our github repo. The results can be found in following files:

Pangenomics: data/pangenome/Pangenomics/CC8_strain_by_gene.pickle.gz

Lines 112 and 315. On what basis did the authors decide on the size of the upstream regulatory region? In the search for mutations, they extracted segments of 300 base pairs, whereas, in the search for the Fur binding motif, only 100 base pairs were considered. The RegPrecise database contains regulons for *Staphylococcus aureus* N315 (https://regprecise.lbl.gov/genome.jsp?genome_id=26), including the Fur regulon with multiple Transcription Factor Binding Sites (TFBSs) that extend beyond the 100 base-pair sequence. I would recommend reconsidering the search within the standardized upstream region of -400 base pairs. In the case of the Fur binding motif search, it might be beneficial to include the TFBSs available in the RegPrecise database.

For Fur motif search, we chose 100 base-pairs because the Fur motif in non-USA300 strains were within ~20 base-pairs of *isdH* translation start site (Figure 4C). In our search of Fur motif in this analysis, we were not looking to see if any exists, we were simply looking to see if the one proximal to the translation start site exists as our DBGWAS analysis suggested that specific region was deleted in USA300 strains.

Line 175. This work aimed to identify potential mutations associated with the success of a specific lineage rather than a phenotype, where correction for population structure effects is necessary. Would the implementation of the bugwas method in DBGWAS for controlling bacterial population structure not potentially impact the results? How was this issue addressed in your analysis? Would it not be pertinent to run a program without population structure correction to enable a comparison of results?

We initially tried to use Linear Mixed Models to find kmers that were only enriched in USA300 strains. These efforts were hampered by extreme linkage disequilibrium which led to high collinearity between kmer abundance making it extremely difficult to get a good estimate of the coefficients. We also tried to run chi-squared tests individually on each kmer which led to unmanageable number (>100k) kmers that were significantly different. DBGWAS on the other hand was able to compress unbranched kmers in the De Bruijn into unitigs and further reduce the number of tests by testing at pattern level instead of unitig level. We found no straight forward way to run DBGWAS (or GEMMA) without population structure correction. Therefore, it is likely we may be underestimating the number of significant unitigs with this approach.

Line 189. Please italicize the gene name cap5E.

Corrected.

Line 277. Please clarify the QC/QA criteria and curation process employed for the selection of RNA-seq experiments, as this constitutes a crucial step in the reconstruction of the network.

We have now added a new supplementary material section, Supplementary Note 2 titled “Creating iModulons for CC8 Clade *Staphylococcus aureus*” with details of QC/QA.

Line 279. In Supplementary Table 3, please label the first column and standardize the use of either the experiment ID or the run ID. Furthermore, verify the experiment identifiers from rows 19 to 26, as I could not locate them in the SRA database.

We have changed all accession to experiment ID including rows 19 to 26.

Lines 290, 330, 424, and 437. Please correct "SCCMec" to "SCCmec IVa" (italicize "mec").

Corrected.

Line 298. What is the size of the upstream regulatory region considered for this analysis? It is important to standardize this value for all analyses involving the upstream regulatory region. In this regard, I recommend maintaining a consistent size of -400 base pairs.

For Fur motif search we chose 100 base-pairs because the Fur motif in non-USA300 strains were within ~20 base-pairs of *isdH* translation start site (Figure 4C). In our search of Fur motif in this analysis, we were not looking to see if any exists, we were simply looking to see if the one proximal to the translation start site exists as our DBGWAS analysis suggested that specific region was deleted in USA300 strains. In our usual analysis, we use -300 base pairs.

Line 321. The discussion is rather concise and lacks an in-depth comparative perspective with relevant literature on any of the obtained results, whether concerning the proposed methodology or the potential new markers associated with the success of the USA300 lineage. The authors must underscore the method is not applicable to all GWAS analyses, due to the issue of correction for population structure.

We have now added sections talking about the importance of *isdH* in *S. aureus* infection and a section addressing the limitation of the current approach when applied to other GWAS type study.

Line 366. The authors employed the methodology described in the article by Hyun et al. 2022 (https://doi.org/10.1186/s12864-021-08223-8) to construct the pangenome. However, this methodology was designed for comparative analysis of pangenomes across various species, which does not align with the objective of this study, focusing solely on *S. aureus* genomes. Consequently, it remains unclear to me why the authors made this particular choice and, more importantly, what advantages it offers over well-established tools for individual pangenomes, such as Roary. I would strongly recommend validating the results using at least one established tool.

With our analysis, we can determine proper thresholds for core/accessory/unique genes based on the observed data (Supplementary Figure 1a). However, we agree that it would be proper to include a more established pangenome package. We have added the results of Roary to our analysis. The Roary results largely agree with our biggest take away from pangenomics which is that our collection of genomes have a good coverage of the CC8 clade at the gene level.

Line 370. Please include the version of CD-HIT that was utilized.

Added. CD-HIT version 4.6 was used for the analysis.

Line 372. What tool did the authors use to extract these regions?

The list of CDS, 5’ and 3’ sequences can be extracted easily with a combination of fasta file and gff file. The gff file was used to find the position of each of these sequences and the sequences were extracted from the fasta file with python scripts.

Line 395. What were the QC/QA criteria used to select the sequences?

The QC/QA criteria for the sequences are mentioned in the beginning of the Pangnomic analysis subsection and is as follows:

“Briefly, “complete” or “WGS” samples from CC8/ST8 were downloaded from the PATRIC database. Sequences with lengths that were not within 3 standard deviations of the mean length or those with more than 100 contigs were filtered out.”

Line 407. Please correct the tool name to "SCCmecFinder" (italicize "mec").

The name has been corrected.

Line 409. I believe BLASTp was run locally, so please specify the version used and the search parameters.

As corrected further down, we used BLASTn not BLASTp. The version v2.2.31 has been added to the methods section.

Line 416. There is conflicting information with line 409, which mentions that PVL was identified through a protein BLAST, but right below, it states it was a BLASTn. Please verify which information is correct and consider the previous comment to specify the version and parameters.

Thank you catching the discrepancy. We have corrected the text:

“PVL was detected using nucleotide BLAST.”

Line 418. Please provide the column identifiers for the Supplementary Table 5 (PVL worksheet).

Column names are added.

Line 418. Please remove the repeated word "and" in Supplementary Table 5 (mecA worksheet) and italicize the gene names in this table.

Corrected

Line 419. You can use the abbreviation "SNPs" since it was introduced in line 65.

Corrected.

Line 420. In my view, this analysis could benefit from a more detailed and clearer explanation.

We have added to the explanation. The section now reads:

“To find the root of the USA300 strains in the phylogenetic tree, the genomes in the tree were first annotated by their PVL and SCC_mec_ status. Then the tree traversed from leaf to root starting from known USA300 strains – TCH1516 and FPR3757- while keeping track of the number of descendant genomes from the current root that contained known markers SCC_mec_ IVa and PVL. The node where the number of genomes with the markers started flatlining was marked as the root of USA300.”

Line 428. Specify the version and parameters used in the analysis with DBGWAS.

Added. The text now reads:

“DBGWAS (v0.5.4) was used to enrich mutations unique to USA300 strains using default kmer size of 31 (-k 31) and neighborhood size of 5 (-nh 5). Alleles with frequency less than 0.1 were filtered (-maf 0.1) and all components enriched with q-values less than 0.05 were documented (-SFF q0.05).”

Line 431. What tools were employed to calculate Pearson correlation and distances relative to the reference genome?

Added. The text now reads:

“Genome-wide linkage was estimated by Pearson correlation (calculated with built-in Pandas function) of the presence/ absence of enriched kmers and distance was measured based on the kmer alignment to the reference TCH1516 genome as determined by BLASTn.”

Line 450. What type of BLAST was used?

Added. Nucleotide blast was used for all kmer analysis.

Line 452. I didn't quite understand the reason for making this analysis available in a separate repository. It would be easier for readers looking to reproduce the work if all the codes were in a single repository.

We kept the repository separate in case we wanted to further develop the network analysis code in the future. We have added the link to the network analysis repository in the README of the publication repo.

Line 460. Please specify the version and parameters, if run locally, or indicate if a web page was used.

Corrected to indicate that we used the PATRIC website for this

Line 470. Specify the version and provide a detailed account of all parameters used, along with the QC/QA criteria and curation methods applied.

We have added Supplementary Note 2 with all the details about packages and parameters used to calculate the iModulons.

Line 479. The phrase "ICA was then run as previously described in chapter 3" does not make sense. Please clarify.

We have corrected the mistake and added a new supplementary note with details about our ICA run. The line now reads:

“A detailed version of the methods for RNA-sequencing and ICA analysis is available as Supplementary Note 2. ICA of RNA sequencing data was performed using the pymodulon package.”

Line 484. Specify the version of CD-HIT.

Added. The version used was v4.6.

Line 494. To enable reproducibility, the repository should be better organized, especially the directory containing the code. Numbering each script in the order it was run would assist the reader in comprehending the overall analysis flow and adapting it to their needs. If creating a manual for method usage is not feasible, the code could be more extensively commented on to explain the parameters, choices made, and how these could be modified. The "Data" folder seems to contain some test files, such as those in the "isdh_fimo" folder, so removing test files would aid the understanding of the reader.

Thank you for the suggestions. We have now numbered the notebooks that generate the figures, we have added more comments to the code, removed testing code and test datasets.

Throughout the article, please correct "SCCMec" to "SCCmec" (italicize "mec").

Corrected.